# Passive acoustic monitoring of baleen whale seasonal presence across the New York Bight

**Bobbi J. Estabrook**[1]*, **Lisa A. Bonacci-Sullivan**[2], **Danielle V. Harris**[1¤a], **Kristin B. Hodge**[1¤b], **Ashakur Rahaman**[1], **Meghan E. Rickard**[3], **Daniel P. Salisbury**[1], **Matthew D. Schlesinger**[4], **Julia M. Zeh**[5], **Susan E. Parks**[5]*, **Aaron N. Rice**[1,6]*

**1** K. Lisa Yang Center for Conservation Bioacoustics, Cornell Lab of Ornithology, Cornell University, Ithaca, New York, United States of America, **2** New York State Department of Environmental Conservation, Kings Park, New York, United States of America, **3** New York Natural Heritage Program, College of Environmental Science and Forestry, State University of New York, Kings Park, New York, United States of America, **4** New York Natural Heritage Program, College of Environmental Science and Forestry, State University of New York, Albany, New York, United States of America, **5** Department of Biology, Syracuse University, Syracuse, New York, United States of America, **6** Department of Public and Ecosystem Health, Cornell University, Ithaca, New York, United States of America

¤a  Current address: Centre for Research into Ecological and Environmental Modelling, The Observatory, University of St. Andrews, Buchanan Gardens, St. Andrews, Fife, United Kingdom
¤b  Current address: Nicholas School of the Environment, Duke University Marine Laboratory, Beaufort, North Carolina, United States of America
* sparks@syr.edu (SEP); arice@cornell.edu (ANR); bobbi.estabrook@cornell.edu (BJE)

**Data Availability Statement:** Whale detection raw data are in S5 Table, and audio survey data are available from the NYS DEC (https://dec.ny.gov/

## Abstract

The New York Bight is an ecologically and economically important marine region along the U.S. Atlantic Coast. Extensive assessments have characterized the habitats and biota in this ecosystem; however, most have focused on fishes, benthic habitats, and human impacts. To investigate the spatial and temporal occurrence of whales in this region, we conducted a three-year passive acoustic monitoring survey that documented the acoustic presence of five baleen whale species that occur within the New York Bight and are of conservation concern: North Atlantic right whales (*Eubalaena glacialis*), humpback whales (*Megaptera novaeangliae*), fin whales (*Balaenoptera physalus*), sei whales (*Balaenoptera borealis*), and blue whales (*Balaenoptera musculus*). Data were recorded with 14 bottom-mounted acoustic sensors across the continental shelf between 2017 and 2020. Right whales were detected across all seasons, with most detections in autumn closer to New York Harbor and spring detections at sites closer to the continental shelf edge. Humpbacks were detected during all months of the year with varying distribution of detections across the shelf. The year-round presence of right and humpback whales challenges previous hypotheses that this region is primarily a stopover location along their migration paths. Fin whales were detected at all sites on most days. Sei whales were detected primarily during the spring at offshore sites. Blue whales were detected in the winter at sites closer to the continental shelf edge, but were rare. These data improve our understanding of baleen whale seasonal occurrences in the New York Bight and can inform monitoring and mitigation efforts associated with the management and conservation of these species.

nature/waterbodies/oceans-estuaries/bight-whale-monitoring-program/passive-acoustic-survey).

**Funding:** Funding for this study was provided by a contract from the New York State Department of Environmental Conservation (https://dec.ny.gov/) to ANR and SEP (Contract C009925). Scientists from NYSDEC (LAB-S, MER) and the NY State Natural Heritage Program (MDS) were involved in survey design and defining project goals, but did not have any influence over the interpretation or presentation of results.

**Competing interests:** The authors have declared that no competing interests exist.

## 1. Introduction

The New York Bight (NY Bight) is an ecologically important marine region within the mid-Atlantic of the U.S. Atlantic Exclusive Economic Zone and has significant environmental and economic value to New York State (NYS) and the United States more broadly. The NY Bight extends along the U.S. East Coast from Long Island and New York City in the north to Delaware Bay in the south, and east to the continental shelf edge. The NY Bight is approximately 12,650 $nm^2$ and includes many ecosystems and taxa. In the 1960s and 1970s, several ecological surveys and assessments were conducted to characterize fish and invertebrate biodiversity and ecosystems in NY Bight [1–7], but these surveys did not include focus on cetaceans. However, relative to fish and invertebrate surveys, there have been few systematic surveys for marine mammals in the NY Bight [1, 2, 5, 7].

Both the U.S. National Oceanic and Atmospheric Administration (NOAA) and the Department of Interior (DOI) recognized the potential importance of the NY Bight to marine mammals and conducted systematic aerial and ship-based visual surveys [8–11]. These surveys included the Cetacean and Turtle Assessment Program that took place between 1979–1980 and had intermittent coverage of the NY Bight [11]. The first systematic passive acoustic monitoring (PAM) surveys of this region were in 2008–2009, when passive acoustic recorders were deployed over three-seasons to characterize baleen whale occurrence [12]. Later, a 2010–2011 PAM survey documented sustained occurrence of endangered North Atlantic right whales (*Eubalaena glacialis*; hereafter referred to as right whales) along the New Jersey Coast [13]. More recently there have been several cetacean survey efforts, including environmental DNA (eDNA) detection [14] and the aggregation of opportunistic visual sightings [15]. A passive acoustic recorder deployed in the Empire Wind Lease Area at the junction of two shipping lanes had a high number of right whale detections in November through March, with occasional detections over the summer, between 2016–2020 [16].

Due to planned expansion of offshore wind development in the NY Bight [17], several remote sensing surveys were initiated to characterize the occurrence of the region's marine protected species. New York State agencies commissioned aerial surveys in 2015 (NYSERDA) and later in 2017 aerial and PAM (NYSDEC) surveys across the NY Bight to systematically characterize cetacean occurrence. The aerial surveys identified the spatial and temporal distributions of large whale species across the NY Bight; fin whales (*Balaenoptera physalus*) and humpback whales (*Megaptera novaeangliae*) were most frequently observed, right whales were sighted in all seasons except summer, and blue whales (*Balaenoptera musculus*) and sei whales (*Balaenoptera borealis*) were rarely seen [18, 19]. Data presented here present passive acoustic survey results of vocalizing baleen whale occurrence across the NY Bight.

Passive acoustic monitoring (PAM) is a particularly useful sensing tool for long-term, continuous monitoring of visually cryptic but highly vocal species, such as baleen whales [20–22]. The primary research output from PAM establishes the spatial and temporal occurrence of focal acoustically active species, and, depending on the survey design and associated corroborating data, PAM can provide more refined estimates of behavior, space use, or density and abundance [e.g., 24–26]. Due to the long-term deployment capabilities of modern recording instrumentation, data can be continuously collected over broad spatial scales, even when other surveys are not possible, including overnight and during inclement weather (e.g., poor visibility, high sea state). PAM requires that an animal is vocalizing, meaning that this survey method can only provide detection and non-detection data rather than true absence data, and cannot detect animals when they are silent [23]. However, the long-term, continuous nature of PAM data allows for an analysis of temporal trends in species detections as well as other ambient noise conditions. Three recent studies aggregated PAM data for baleen whales across the U.S.

Atlantic coast and synthesized the current scientific understanding of baleen whale occurrence, migration, and ecology across the Atlantic [see 26–28]. Davis et al. [26, 27] identified changes in the distribution of baleen whales before and after 2010, but these meta-analyses only included archival acoustic data through 2014. PAM is also a key strategy in monitoring cetaceans in offshore wind areas along the US Atlantic Coast and mitigating impacts from renewable energy development [29].

The combination of anticipated expanding human use of NY Bight and species range shifts due to climate change raise fundamental research and management questions as to where and when do large whale species occur within NY Bight; these data become important for updating protected species management approaches needed for effective conservation measures.

As part of the NYSDEC New York Bight Whale Monitoring Program (https://www.dec.ny.gov/lands/113647.html), we conducted a PAM survey from October 2017 to October 2020 in the NY Bight to investigate temporal and spatial trends in acoustic detections of five baleen whale species: North Atlantic right whales (*Eubalaena glacialis*), humpback whales (*Megaptera novaeangliae*), fin whales (*Balaenoptera physalus*), sei whales (*Balaenoptera borealis*), and blue whales (*Balaenoptera musculus musculus*). These taxa are identified as "Species of Greatest Conservation Need" by New York State [30], and are the likely mysticetes to be found on the continental shelf within New York Bight. These focal species have been documented to occur across NY Bight, but is unclear how these species use this area as a habitat, whether it is part of their north-south migratory corridor (in the case of right whales and humpback whales), or whether it at the edge of their geographic range extent (as for sei whales and blue whales) [26, 27]. These data will be used to inform a larger scale monitoring plan for guiding state and federal level management efforts for mitigating effects of expanded anthropogenic activities (in particular offshore wind) across the NY Bight on marine protected species, and can reveal broader temporal presence trends of the five baleen whale species in the NY Bight.

## 2. Materials and methods

### 2.1 Data collection

Passive acoustic data were collected using two different bottom-moored archival digital acoustic recording devices: Marine Autonomous Recording Units [MARUs; 31], and Autonomous Multichannel Acoustic Recorders (AMARs, http://www.jasco.com/amar). The MARUs were suspended 2 m from the seafloor and recorded continuously with a sample rate of 5 kHz at 12-bit resolution, a high-pass filter at 10 Hz, and a low-pass filter at 2 kHz. The effective recording bandwidth (10–2000 Hz) had a sensitivity of -168 dB ± 3.0 dB (re: 1V/μPa) with a flat frequency response between 15 and 585 Hz. Sites for which MARUs were used are named with "M" in their site-specific names. AMARs were suspended 1 m from the seafloor and recorded continuously at 8 kHz with a 24-bit resolution. The AMAR hydrophones were calibrated with a sensitivity of -164 dB (re: 1V/μPa at 1 kHz). Sites for which AMARs were used contain "A" in the site name. AMARs were deployed to characterize higher frequency noise levels along the southbound shipping lanes out of NY harbor.

Fourteen sites were configured across two transects along the Nantucket-Ambrose and Ambrose-Hudson Traffic Separation Schemes (Fig 1) to capture the sounds of cetaceans moving across the continental shelf near the primary NY Harbor shipping lanes. Data were collected from 16 October 2017 through 15 October 2020, with occasional gaps in the recording periods when units were trawled (i.e., dragged up by fishing vessels), lost (not recovered), or malfunctioned (Table 1, Fig 2, S1 Table).

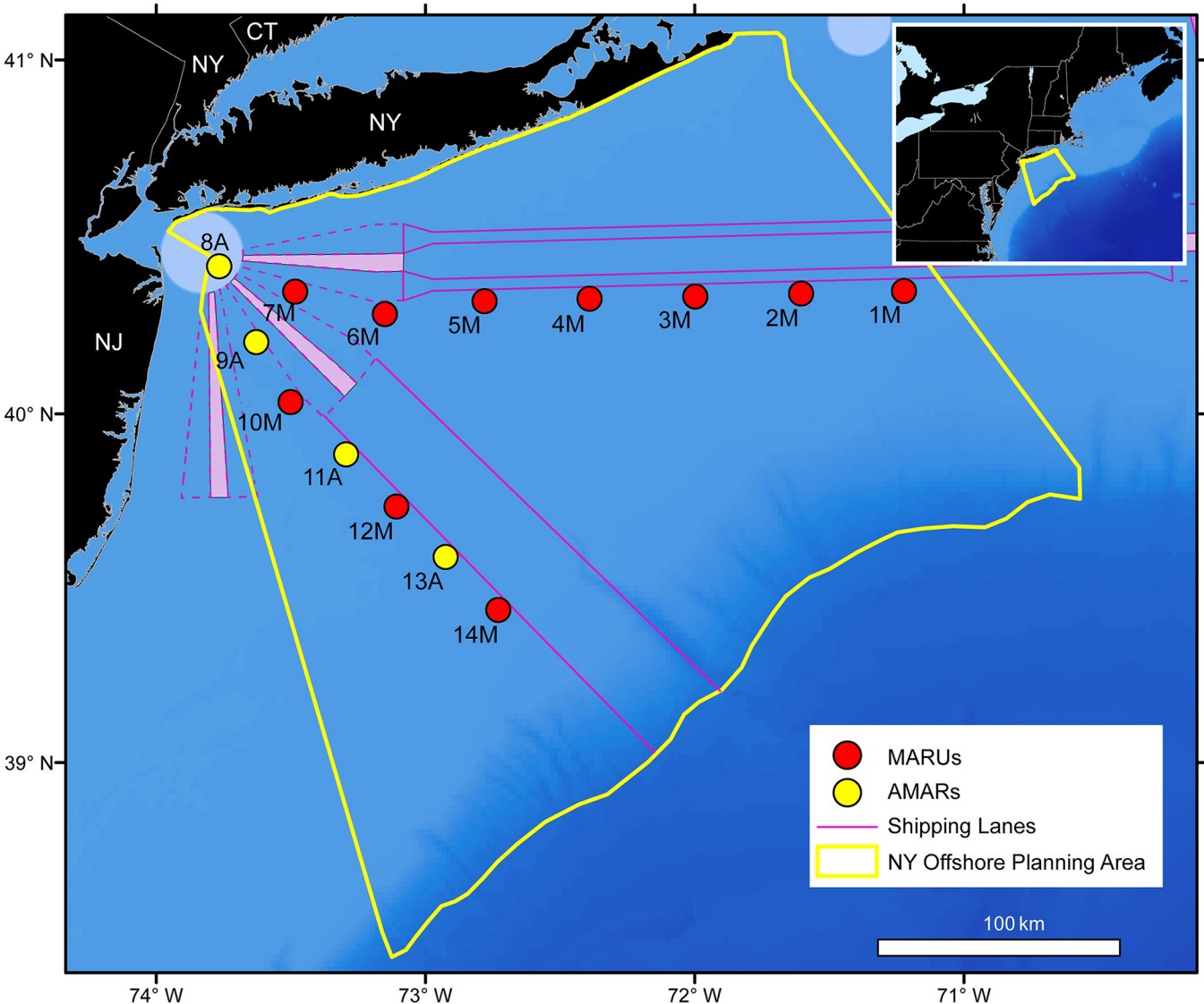

**Fig 1. Map of the acoustic recording locations within the New York Bight, with sensor location numbers.** "M" denotes MARUs (red), and "A" denotes AMARs (yellow). Inset shows the New York Bight at a larger spatial scale for geographical context. Shipping Areas to be Avoided represent shipping lanes and turning basins for vessel traffic in the NY Bight. State boundary maps are from ArcGIS, Shipping Lane Maps are from marinecadastre.gov, and bathymetry data are from GEBCO.

## 2.2 Acoustic detection of focal whale species

We used a combination of custom automated detection algorithms with human-aided visual and aural detection to determine the daily acoustic presence of right, humpback, fin, sei, and blue whales in the NY Bight (species-specific analysis approaches are described in more detail below). Here, "presence" refers to positive acoustic detection events of a focal species in the audio files. A lack of acoustic detection does not imply absence, rather that focal species vocalizations were not detected at that time. Automated detectors were used to identify right, fin, and sei whale signals, and each detection was reviewed and validated by an analyst to confirm true positive detections for each day and site. Manual detection was used to identify humpback and blue whale signals, for which spectrograms of audio files were reviewed for each date and

**Table 1. Deployment location, depth, start date of first deployment, end date of last deployment, and total recorded days of passive acoustic surveys at each site in the New York Bight, 2017–2020.**

| Site | Latitude (°N) | Longitude (°W) | Depth (m) | Start Date | End Date | Total Recording Days |
|------|---------------|----------------|-----------|------------|----------|----------------------|
| 1M | 40.348 | -71.224 | 90 | 2017-10-16 | 2020-10-15 | 1031 |
| 2M | 40.342 | -71.606 | 84 | 2017-10-16 | 2020-07-09 | 679 |
| 3M | 40.334 | -71.999 | 65 | 2017-10-16 | 2020-10-15 | 848 |
| 4M | 40.327 | -72.408 | 54 | 2017-10-16 | 2020-10-05 | 693 |
| 5M | 40.320 | -72.782 | 50 | 2017-10-16 | 2020-06-20 | 655 |
| 6M | 40.284 | -73.152 | 40 | 2017-10-16 | 2020-06-20 | 727 |
| 7M | 40.347 | -73.484 | 30 | 2017-10-16 | 2020-01-21 | 757 |
| 8A | 40.414 | -73.762 | 28 | 2017-10-16 | 2019-10-15 | 492 |
| 9A | 40.200 | -73.628 | 38 | 2017-10-16 | 2019-10-15 | 733 |
| 10M | 40.033 | -73.500 | 49 | 2017-10-16 | 2020-10-15 | 856 |
| 11A | 39.883 | -73.292 | 49 | 2017-10-16 | 2020-01-10 | 814 |
| 12M | 39.734 | -73.106 | 51 | 2017-10-16 | 2020-10-15 | 896 |
| 13A | 39.589 | -72.922 | 63 | 2017-10-16 | 2019-10-19 | 732 |
| 14M | 39.440 | -72.730 | 80 | 2018-04-12 | 2020-10-15 | 801 |

site. Audio files and detections were viewed as spectrograms using Raven Pro 2.0 Sound Analysis Software [32]. All confirmed target signals were reviewed again by a second, expert human analyst to improve data accuracy by eliminating false detections.

To evaluate daily acoustic presence of right whales at each site, we focused on right whale upcalls. Upcalls are one of the most common sounds produced by North Atlantic right whales [33–36], and are frequently used to determine right whale acoustic presence [e.g., 12, 37–39]. Upcalls are produced by males and females of all age classes and appear to be produced in

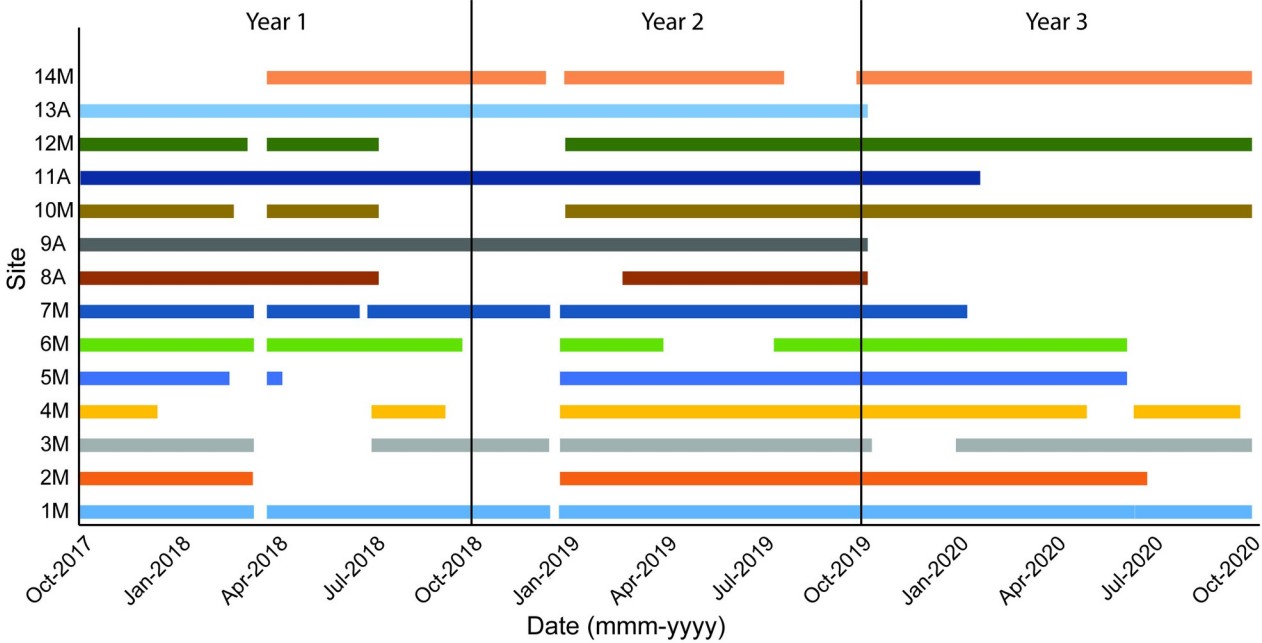

**Fig 2. Recording effort for each MARU and AMAR site during the three survey years (16 October 2017–15 October 2020).** The multi-colored horizontal bars represent each site and indicate the time periods in which acoustic data were recorded. The white gaps along the site lines indicate times when acoustic data were not recorded (i.e. data gaps).

social contexts [40–42]. They are characterized as a frequency-modulated upsweep with a typical duration between 0.3 and 1.5 s within a frequency band of approximately 71–224 Hz [43], with a bandwidth of 100 ± 37 Hz SD [44]. A MATLAB-based (MathWorks, Natick, MA) automated detection algorithm was used to detect upcalls [45]. Spectrogram settings for reviewing detections included a 60-s page duration, a frequency range of 10–450 Hz, and a fast Fourier transform (FFT) window size of 512 points for the 5 kHz MARU audio files and 2048 points for the 8 kHz AMAR files. Automated detection of right whale upcalls sometimes falsely detects humpback whale signals with similar acoustic properties [41, 46]. To prevent these types of false positives, upcall detections with concurrent humpback whale signals were not included in the right whale presence analysis if the detection could not be distinguished from humpback sounds.

Humpback whale songs and social calls (non-song vocalizations) were used to determine daily humpback whale acoustic presence. Humpback whale songs have been recorded only from males, whereas social calls are produced by both sexes and all age classes in various of contexts[e.g., 47, 48]. Thus, song and social call detections account for male and female presence. We manually reviewed 1 of every 4 h of audio for humpback signals. When tested using a subset of 50 days, a review of 25% of the audio files yielded the same daily presence results for humpback whales as manually reviewing 100% of the audio files. Spectrogram settings included a 5-min page duration, frequency range of 10–600 Hz, and a FFT window size of 512 points for MARU audio files and 1024 points for AMAR files.

Fin whales produce a song composed of long sequences of short individual notes centered around 20 Hz, and song bouts can last up to 32 h [49–51]. Songs are hypothesized to be produced only by males in feeding and mating contexts [52, 53]. Since the 20-Hz notes have been recorded in parts of the western North Atlantic throughout most of the year and over long durations, they are likely to be recorded while male fin whales are present in the NY Bight, and are, therefore, good indicators of male fin whale presence. We used a MATLAB-based matched-filter data-template detector algorithm [54] to automatically detect daily presence 20-Hz notes in the audio files [55]. Down-sampled 400-Hz audio files were used for this analysis, with spectrogram settings of a 90-s page duration, 0–100 Hz frequency band, and an FFT window size of 512 points.

Daily presence of sei whales was determined by using the template detector in Raven Pro 2.0 Sound Analysis Software [32] to detect sei whale downsweeps, which are low-frequency (34–82 Hz) signals of approximately 1.4 s [56], and can occur singularly, in doublets, or triplets [57]. It is not known if downsweeps are sex-specific or which behavioral context they are associated with. Six representative sei whale downsweep examples were selected from our dataset and used as templates for cross-correlation, and then applied as a classifier to the complete audio data set to find instances of sei whale downsweeps. Spectrogram settings comprised a 60-s page duration, 0–200 Hz frequency band, and an FFT window size of 1024 points for MARU audio files and 2048 points for AMAR audio files. We observed many distorted, mode-dispersed sei whale downsweeps (S1 Fig) that might be due to long-distance signal propagation in the shallow water of the NY Bight [58–60].

Blue whale daily presence was determined by manual review for characteristic patterns of blue whale 14–22 Hz sounds. To increase computational efficiency involved in analysis, audio files were decimated (i.e. frequency range was reduced) to 200 Hz. Blue whales in the North Atlantic produce song that is characterized by a sequence of phrases, most commonly composed of part A, a tonal signal, and part B, a descending tonal signal (referred to as the A-B phrase), between 15 and 20 Hz [61]. Only male blue whales have been confirmed as singers [62]. Spectrogram settings included a page length of 45 min and a frequency range of 10–25 Hz, with an FFT window size set to 1025 points.

## 2.3 Detector performance evaluation

Detector performance was evaluated using a subset of data that spanned the 3-year survey period. Beginning with a random date (24 October 2017) through the end of the survey, we manually searched audio files for each instance of the target signals on every 20[th] day, where between 1 and 3 sites were checked for every selected day, yielding a human-validated, groundtruthed dataset with 58 calendar days. Right whale acoustic presence was sparse in the groundtruthed dataset, so we reviewed every 10[th] day during months when right whales were commonly detected in the NY Bight (November–May), yielding 80 calendar days (171 site-days) for the right whale groundtruthed dataset. The manually annotated groundtruthed data-sets were compared against the detector output to determine true positive rates (TPR) on a daily scale. Days in which at least one automated detection overlapped in time and frequency with a groundtruthed signal were considered true positive (TP) days. Days in which there was at least one groundtruthed signal but no overlapping detections were defined as false negative (FN) days. The true positive rate (TPR) was the sum of TP days divided by the sum of the TP days and FN days for the corresponding target signal.

The TPR of the right whale upcall detector was 0.69, where 20 of the 29 days with presence had at least one TP detection and 9 days had FN. The fin whale detector had a TPR of 0.99, with 85 TP days and 1 FN day. The sei whale detector had a TPR of 0.33, with 7 TP days and 12 FN days. The detector had a lower TPR for sei whale downsweeps with multi-modal disper-sion than for downsweeps that did not exhibit significant modal dispersion, likely because mode-dispersed downsweeps were not included in the template set. This suggests that down-sweeps that originated near an acoustic sensor were more likely to be detected, as those signals are less susceptible to multi-modal dispersion. The effect of the distorted sei whale signals on the downsweep template detector performance is described further in the Supporting Information.

## 2.4 Statistical analysis of whale acoustic presence

To describe seasonal acoustic presence of each focal species, we used generalized additive models (GAM). A GAM allows for non-linear relationships between the response and predic-tor variables by using smoothed functions, providing more flexibility when modeling temporal trends. With these models, we tested if there was monthly variation in species presence and if species presence differed between survey years. Models were fitted using the 'bam' function of the 'mgcv' package in R [63], and were structured with detection evets per day at each site as a Bernoulli response variable (detection/no detection) using a binomial distribution with a logit link function. Predictor variables included month as an integer with a cyclic smooth function and survey year as a factor. We defined survey years as follows: Year 1 (October 16, 2017 – October 15, 2018), Year 2 (October 16, 2018- October 15, 2019), and Year 3 (October 16, 2019 –October 15, 2020). Site as a factor was included as a random effect. To address temporal auto-correlation from daily samples, a continuous-time first-order autoregressive correlation struc-ture (AR1) was included in the model. Models were fitted separately for each species using the same parameters between models. To compare the goodness of fit and model complexity of the GAM with the AR1 and the GAM without the AR1, we compared the Akaike information criterion (AIC), where the model with the lowest AIC and a difference > 2 was selected as the better fitting model. Model checking was conducted using the 'DHARMa' package [64] in R to confirm that model assumptions (e.g., homoscedasticity, independence, normality, dispersion, and zero-inflation) were not violated, following Zuur et al. 2009 [65]. To make the results of the GAM more interpretable, we back-transformed model estimates using the 'emmeans' package in R [66] and present the estimated marginal means (EMM) of the response variable

**Table 2. Referenced signal type, frequency band, source levels and source depth of the target species used for ambient noise calculations and detection range estimation.**

| Species | Target Signal | Frequency Band | 1/3$^{rd}$-Octave Frequency Band | Source Level (dB re 1μPa @ 1m RMS) ± SD | Source Depth |
|---|---|---|---|---|---|
| **North Atlantic Right Whale** | Upcall | 71–224 Hz [43] | 70.8–224 Hz | 172 ± 6.6 [34, 67] | 2 m [42] |
| **Humpback Whale** | Song | 29–2480 Hz (Dunlop et al. 2007) | 17.8–708 Hz | 169 ± 3 [68] | 50 m [69] |
| **Fin Whale** | 20-Hz pulse | 17–25 Hz [51] | 17.8–28.2 Hz | 189 ± 4 [70, 71] | 20 m [72] |
| **Sei Whale** | Downsweep | 34–82 Hz [56] | 28.2–89.2 Hz | 173.5 ± 3.2 [73] | 20 m [58] |
| **Blue Whale** | Song | 15–19 Hz [61] | 14.1–22.4 Hz | 189 ± 3 [74] | 30 m [75] |

on the response scale. The response variable represents the proportion of time with focal whale presence per month and year. A statistical pair-wise comparison of the estimated marginal means between each survey year was conducted using the Tukey method on the log odds ratio scale.

## 2.5 Detection range estimation

Each of the focal whale species' signals has a different detection distance, which can be affected by habitat features and noise conditions in the environment. To better understand the potential area of coverage for this survey, we estimated approximate feasible detection ranges of each species in the NY Bight using an assumed source level of each target signal (Table 2), the ambient noise levels in the target signal frequency band, and physical and oceanographic factors in the study area that can influence sound propagation of the signal.

Ambient noise levels were quantified as equivalent continuous sound pressure level, $L_{eq}$ (dB$_{rms}$ re 1 μPa) [e.g., see 76], and measured from the audio data using SEDNA [77] in MATLAB, with a Hann window with zero overlap, FFT with a resolution of 1 s and 1 Hz. The $L_{eq}$ represents the average unweighted sound level of a continuous time-varying pressure signal over intervals of 10 min, a temporal resolution that captures changes in noise levels from a passing vessel [67], a dominant sound source in the NY Bight. We used 1/3$^{rd}$-octave species-specific frequency bands (Table 2) to represent the frequency range in which each species' target signals occur [28, 47, 67, 70]. We also calculated noise levels across the full frequency bandwidth (based on center frequencies of 1/3$^{rd}$-octave bands. 8.9–2239.6 Hz) to describe broadband noise conditions during the survey period.

To estimate plausible detection ranges for each species in the NY Bight during this survey, we calculated the received levels of each target signal at varying distances from the receiver (i.e., acoustic sensor) given ambient noise levels at the receiver, following the Acoustic Integration Model-based methods [78] of calculating receiver communication space [see 43, eq. 10a-c]. we used the following one-way transmission loss (TL) model:

$$TL = 20log_{10}(H) + 17log_{10}\left(\frac{R}{H}\right) \tag{1}$$

where $H$ is the depth of the source, and $R$ represents the range of the source to the receiver. The intermediate (relative to cylindrical and spherical spreading) $17log_{10}R$ spreading loss model [67, 79] was applied to the custom MATLAB package to estimate sound propagation. The calculated TL was then used to estimate the received level (RL) of the target signal when it reaches the receiver using the passive sonar equation: RL = SL–TL. The estimated detection range is the distance from the source at which the target signal is detectable over ambient

noise, and therefore the signal-to-noise ratio equals 0. The model incorporates bathymetry in its propagation calculation by using spherical spreading at the approximate water depth and intermediate spreading beyond that calculated threshold [e.g., 80, 81]. We estimated the detection range of each whale species at specific range steps for eight different bearings relative to the sensor (0˚, 45˚, 90˚, 135˚, 180˚, 225˚, 270˚, 315˚) at each site location to account for varying bathymetry around each sensor, during high (95th percentile), median (50th percentile), and low (5th percentile) noise conditions. We then averaged the range estimates across bearings to estimate a single detection range value for each species and noise level percentile. Target species source depths, source levels, and frequency bands used in the models are presented in Table 2. The detection range estimates presented here do not illustrate temporal variation as a function of noise levels on the seasonal or annual scale, but rather overall noise levels during the survey. Therefore, we present seasonal and annual $L_{eq}$ statistics in the Supporting Information to reflect potential temporal variation in relative detection range estimates. Moreover, many variables are not included in this model, and some variables that are included are inherently variable themselves (e.g., source level), therefore these estimates are approximate.

## 3. Results

### 3.1 Acoustic presence of focal whale species

During the survey period (16 October 2017–15 October 2020), 1095 calendar days of audio data were collected, during which all 5 baleen whale species were acoustically detected with varying seasonal and spatial distributions across the NY Bight (Figs 3 and 4, Table 3). For each species, the best fitting GAM was the model that incorporated the AR1 structure, and the model assumptions were met.

**3.1.1 North Atlantic right whale presence.** North Atlantic right whale upcalls were detected on 42% (458) of the days (Table 3), during each month of the year, with a peak in detections during winter months (November–January), a small secondary peak in spring months (March–May), and the fewest detections in July through October (Fig 3A, Tables 3 and S2). Right whales were detected at all recording sites (Fig 4A), with the fewest detections at site 8A (3% of days) and most detections at sites 9A and 11A (14% and 15% of days) (S3 Table). Right whales demonstrated strong seasonal spatial patterns, with more detections at sites closer to the continental shelf edge from March to May, and at sites nearer to NY Harbor from November to February (S2 Fig). There was a significant effect of survey year on daily detections (GAM: $X^2$ = 16.72, d.f. = 2, $p$ < .001), where the probability of upcall detections occurring during Year 1 (EMM = 0.102, SE = 0.017, d.f. = 10688) and Year 2 (EMM = 0.106, SE = 0.015, d.f. = 10688) were significantly different ($p$ < .005) from Year 3 (EMM = 0.038, SE = 0.009, d.f. = 10688), while Year 1 and Year 2 were not significantly different from each other (Tukey adjustment $p$ value = 0.970) (Fig 3B). During Year 1 and Year 2, right whales were detected most at sites 9A, 11A, and 13A, comprising approximately 27% and 31% of the daily upcall detections at those sites during each year, respectively. However, the sensors at those sites experienced substantial data loss during Year 3 (Fig 2), limiting our interpretation of these interannual differences in presence.

**3.1.2 Humpback whale presence.** Humpback whale vocalizations were detected on 79% (865) of the recording days during the survey period, and in all months of the year, with variable presence across seasons (Fig 3C, Tables 3 and S2). Humpback whales were detected across all sites, with the fewest detections at site 8A (9% of recorded days), while daily detections at other sites ranged between 21% (site 2M) and 53% (site 13A) of the recording days (Fig 4B, S3 Table). Humpback whales demonstrated seasonal spatial patterns, where more detections occurred at sites nearer to NY Harbor between November and March, and at sites nearer to

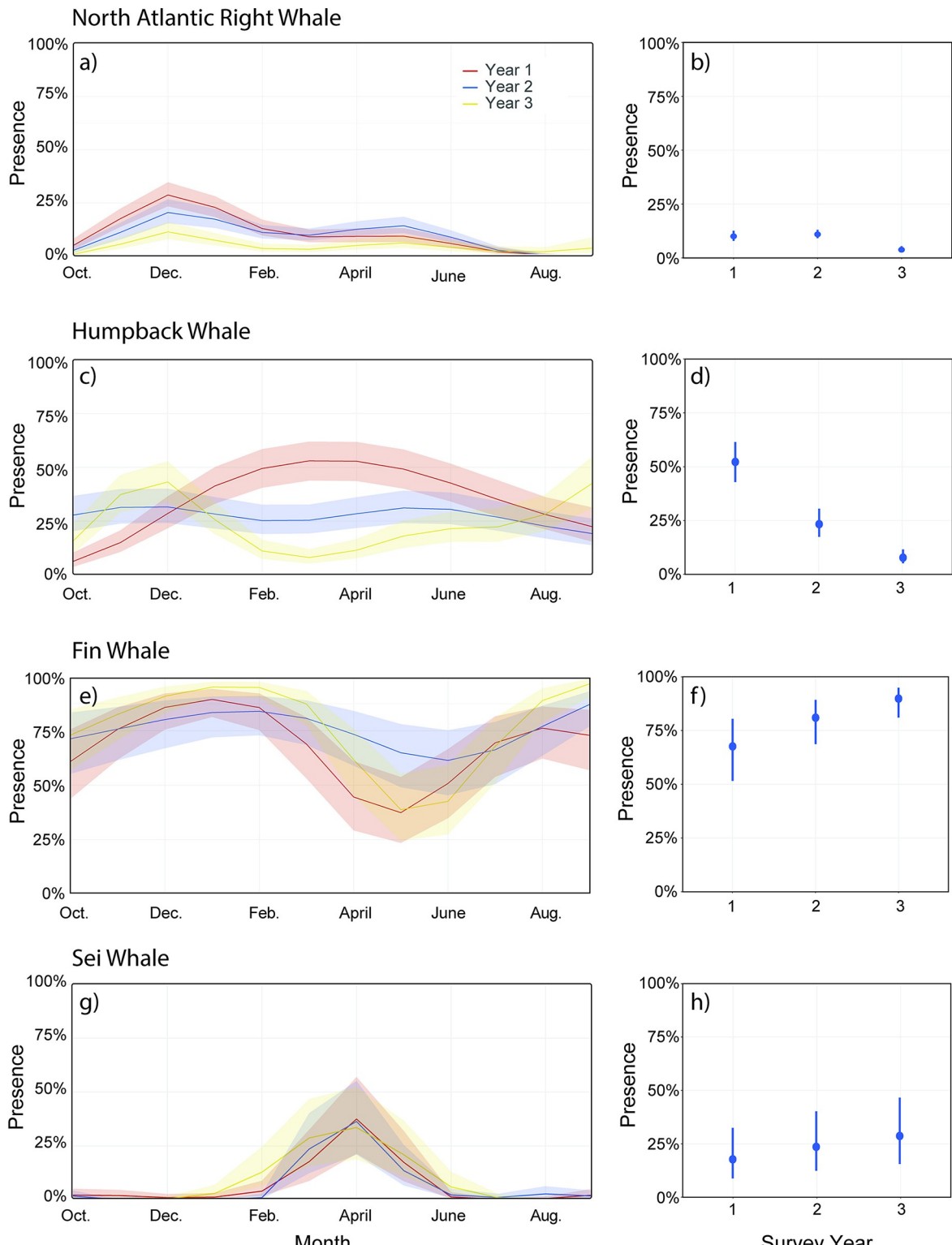

**Fig 3.** GAM plots of daily acoustic presence of right whales, humpback whales, fin whales and sei whales (panels a, c, g, e), with the estimated marginal means of the probability of presence on the y-axis by month and survey year (panels b, d, f, h). For panels a, c, g, e, Survey Year 1 (October 2017–2018) is indicated by a red line, Year 2 (October 2018–2019) is indicated by a blue line, and Year 3 (October 2019–2020) is indicated by a yellow line.

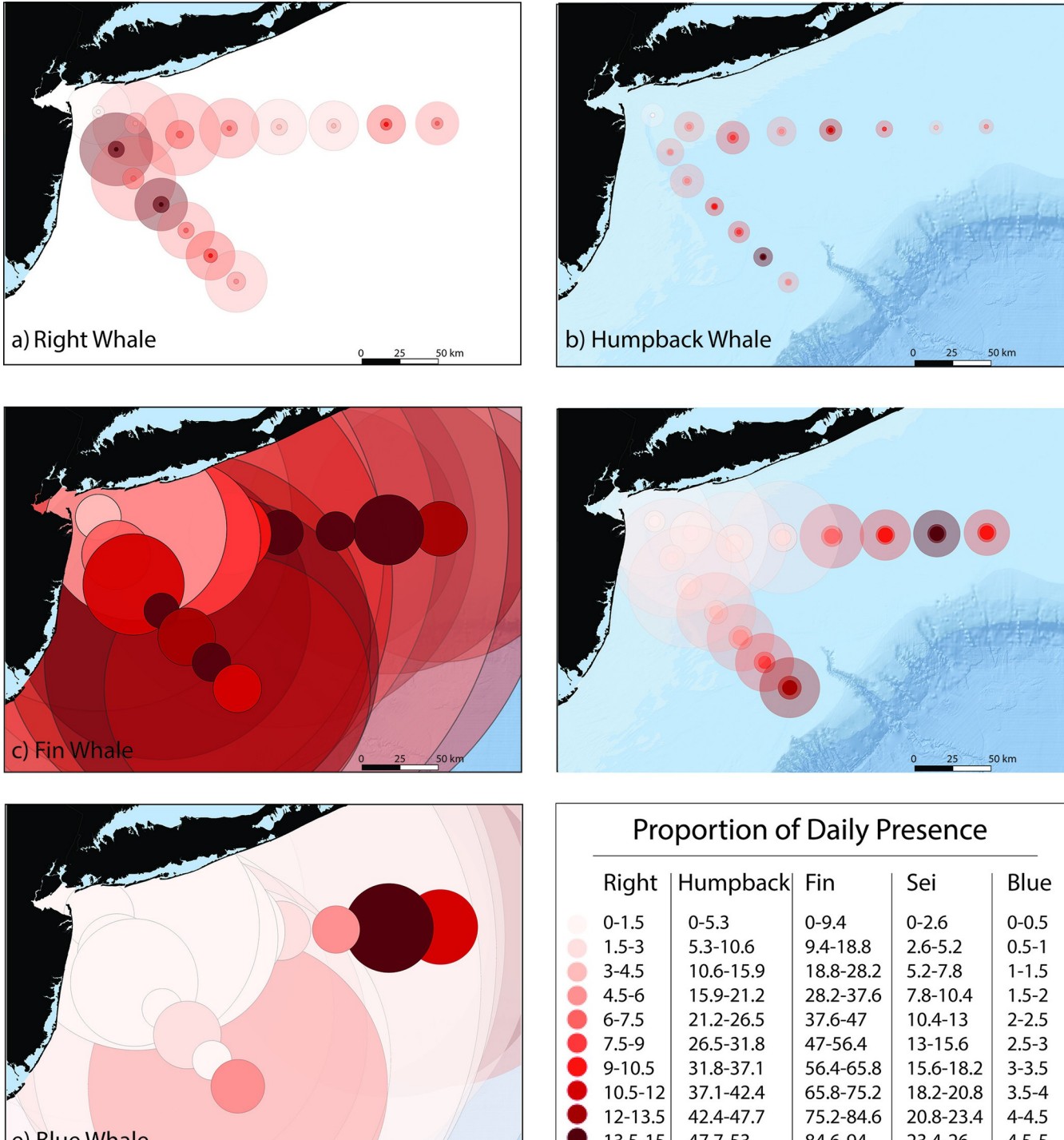

**Fig 4. Detection area and cumulative spatial distribution of focal whale species detected between 16 October 2017 and 15 October 2020.** The size of the circles around each site represents the estimated detection area based on the 5th, 50th, and 95th noise percentiles (the 5th percentile detection areas were excluded from the fin whale and blue whale plots because they encompassed the entire map area). The colormap is scaled to the proportion of days with detections for each species over total days recorded, where white represents 0 presence and deep red represents the highest presence for each species. State boundary maps are from ArcGIS and bathymetry data are from GEBCO.

**Table 3. Daily acoustic detections per species by season and year in the New York Bight during Fall: September–November, Winter: December–February, Spring: March–May, and Summer: June—August.** 'Total Days Recorded' is the total calendar days recorded, '# Days Detected' refers to the number of calendar days with detections. '% Days Detected' is the percentage of recording days with a detection of the species.

| Season | Total Days | Right | | Humpback | | Fin | | Sei | | Blue | |
|---|---|---|---|---|---|---|---|---|---|---|---|
| | | # Days Detected | % Days Detected | # Days Detected | % Days Detected | # Days Detected | % Days Detected | # Days Detected | % Days Detected | # Days Detected | % Days Detected |
| Fall 2017 | 46 | 27 | 59% | 14 | 30% | 46 | 100% | 17 | 37% | 3 | 7% |
| Winter 2017–2018 | 90 | 66 | 73% | 82 | 91% | 90 | 100% | 22 | 24% | 23 | 26% |
| Spring 2018 | 92 | 55 | 60% | 79 | 86% | 90 | 98% | 81 | 88% | 0 | 0% |
| Summer 2018 | 91 | 14 | 15% | 88 | 96% | 91 | 99% | 15 | 16% | 0 | 0% |
| Fall 2018 | 91 | 26 | 29% | 85 | 92% | 91 | 100% | 8 | 12% | 0 | 0% |
| Winter 2018–2019 | 90 | 68 | 76% | 72 | 80% | 90 | 100% | 2 | 2% | 7 | 8% |
| Spring 2019 | 92 | 77 | 84% | 88 | 96% | 92 | 100% | 88 | 96% | 0 | 0% |
| Summer 2019 | 92 | 32 | 35% | 86 | 93% | 92 | 100% | 24 | 26% | 0 | 0% |
| Fall 2019 | 91 | 17 | 19% | 78 | 86% | 91 | 100% | 7 | 8% | 0 | 0% |
| Winter 2019–2020 | 91 | 33 | 36% | 64 | 70% | 88 | 97% | 31 | 34% | 17 | 19% |
| Spring 2020 | 92 | 23 | 25% | 36 | 39% | 92 | 100% | 82 | 89% | 0 | 0% |
| Summer 2020 | 92 | 15 | 16% | 50 | 54% | 90 | 98% | 16 | 17% | 0 | 0% |
| Fall 2020 | 45 | 5 | 11% | 43 | 96% | 45 | 100% | 0 | 0% | 0 | 0% |
| **Total** | 1095 | 458 | 42% | 865 | 79% | 1088 | 99% | 393 | 36% | 50 | 5% |

the shelf edge between July and September (S3 Fig). Survey year had a significant effect on daily humpback whale detections (GAM: $X^2$ = 34.84, d.f. = 2, p <001), and each survey year was significantly different (Tukey adjusted p value < .001) from the others, where Year 1 (EEM = 0.522, SE = 0.047, d.f. = 10687) had the highest probability of detections, followed by Year 2 (EEM = 0.252, SE = 0.034, d.f. = 10687), and Year 3 (EEM = 0.090, SE = 0.018, d.f. = 10687) (Fig 3D). However, the sites missing from the Year 3 survey (9A, 11A, and 13A) accounted for 12% and 10% of all daily detections during Year 1 and Year 2.

**3.1.3 Fin whale presence.** Fin whale 20-Hz pulses were the most frequently detected among baleen whale signals, with detections on 99% (1088 days) of the recording days (Tables 3 and S2). While prominent throughout the survey, fin whales exhibited slight seasonal trends in presence, with the fewest detections between April and June (Figs 3E and S4). Fin whales were detected >50% of the days at most sites, with the fewest daily detections at sites near NY Harbor and more detections at sites closer to the continental shelf edge (Figs 4C and S4), demonstrating some spatial variability of fin whale detections throughout the NY Bight. Survey year was a significant predictor of daily detections at all sites ($X^2$ = 44.21, d.f. = 2, p < 0.001), where the probability of daily detections during Year 1 (EMM = 0.546, SE = 0.084, d.f. = 10687) was significantly lower (p < 0.001) than Year 2 (EMM = 0.776, SE = 0.058, d.f. = 10687) and Year 3 (EMM = 0.798, SE = 0.056, d.f. = 10687), indicating interannual differences in presence (Fig 3F).

**3.1.4 Sei whale presence.** Sei whale downsweeps were detected during all months of the year and on 36% (393 days) of the recorded days. Sei whale detections exhibited strong seasonality where a single peak in daily detections occurred during spring months, between March and May (Figs 3G S5, Tables 3 and S2) in all three survey years. Sei whales were detected

most at sites near the continental shelf edge (Sites 2M: 26% days; 14M: 23%; 1M: 19%) and the least at sites closer to NY Harbor (Figs 4D and S5), with daily detections ranging between 1% (sites 7M and 8A) and 5% (sites 5M, 6M, and 9A) of the recording days. The probability of daily sei whale detections in the NY Bight were not significantly different (p > 0.05) between Year 1 (EMM = 0.286, SE = 0.082, d.f. = 10687), Year 2 (EMM = 0.353, SE = 0.090, d.f. = 10687), and Year 3 (EMM = 0.330, SE = 0.870, d.f. = 10687), indicating low interannual variability in sei whale detections in the NY Bight during this survey (Fig 3H).

**3.1.5 Blue whale presence.** Blue whales were the least frequently detected species (Tables 3 and S2), with acoustic detections on 5% (50) of the recorded days. Blue whales were detected between November and February only, and only at sites near the continental shelf edge (Figs 4E and S6). Sites 1M and 2M recorded the most days with blue whale detections, where 4% (42 days) and 5% (36 days) of the recorded days had detections, respectively, while detections at the remaining sites occurred on 0%– 2% of the recorded days. Since there were very few days with detected blue whale song, it was not appropriate to test for temporal trends statistically. Daily detections varied little between Year 1 (26 days, 7%), Year 2 (7 days, 2%), and Year 3 (18 days, 5%).

## 3.2 Detection range estimation

**3.2.1 Ambient noise measurements.** Noise levels varied across the NY Bight (Fig 5), where site 8A recorded the highest median $L_{eq}$ within the right whale, humpback whale, and sei whale frequency bands, limiting the range of detectability for those species' target signals. In all frequency bands, median $L_{eq}$ were generally lower at sites 4M – 7M, 9A, and 10M, while sites nearer to the shelf edge recorded slightly higher $L_{eq}$ values. The full and humpback whale frequency bands, the broadest frequency bands measured here, recorded the overall highest $L_{eq}$, followed by the sei and right whale frequency bands (Fig 5). Overall, noise levels were lowest during summer (June—August) and highest during fall (September—November) and winter (December—February) (S7–S12 Figs) for all frequency bands.

**3.2.2 Detection range by species.** During median noise conditions, right whale upcalls were estimated to have an average detection range of 7 km (SE = 0.38) across all sites and bearings for a source level of 172 dB re 1μPa. Median humpback whale song detection range estimates were 3 km (SE = 0.14) for a source level of 169 dB. Humpback whale social calls, which were also detected in this study, are produced at variable source levels. They have lower source levels than song [164 dB for social songs vs. 170 dB for song; described in 82] and would therefore have a smaller detection range. Sei whale detection range estimates were approximately 11 km (SE = 0.63) for 173.5 dB source level. Fin and blue whale detection ranges were longest, with estimated ranges exceeding 160 km for both species, based on median noise levels within their respective frequency bands (SE = 16 and 12 km, respectively). Detection range estimates for each noise percentile and source level per species' frequency band can be found in S5 Table.

## 4. Discussion

These results represent a comprehensive, multi-year continuous PAM survey of right, humpback, fin, sei, and blue whale presence in the NY Bight. Right, fin whales and humpback whales were detected in the NY Bight throughout most of the year, strengthening the evidence [19, 83] that this region serves as more than just the migratory corridor that was previously suggested [84–87]. Baleen whale detections were distributed broadly across the NY Bight. Fin, sei, and blue whales were detected primarily at sites closer to the shelf edge, while right and humpback whales were detected more broadly across the shelf (Fig 4).

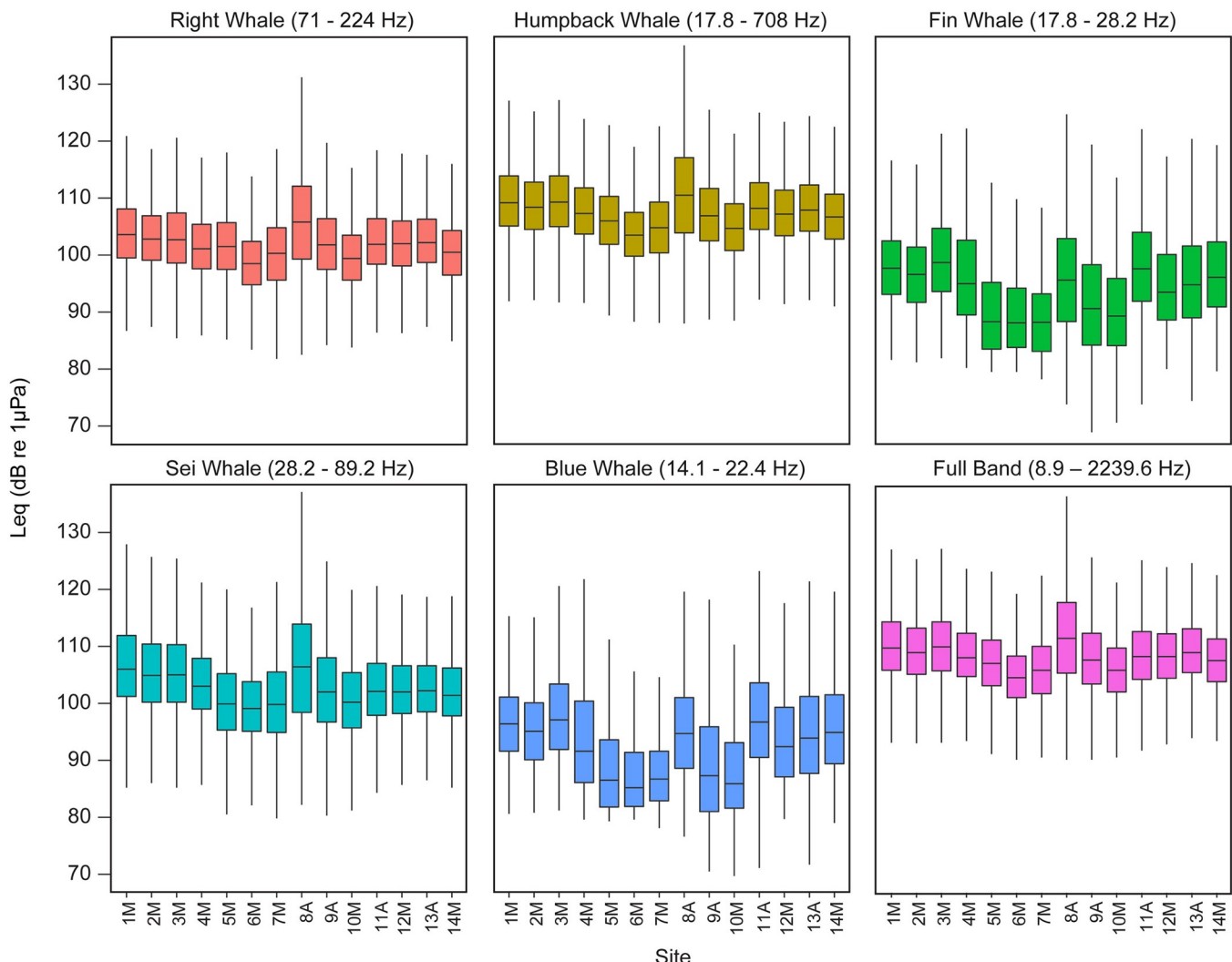

**Fig 5. Boxplots of 10-minute averaged Leq (dB re 1µPa) per site for each frequency band between 16 October 2017 and 15 October 2020.**

The year-round temporal occurrence of baleen whale species in the NY Bight raises several interesting questions about how the region fits in to the life history of these species and has direct implications for management. The year-round occurrence of right whales and humpback whales in the NY Bight challenges previous hypotheses that this region is primarily a stopover location as part of their migration paths. Additionally, it is unclear how most baleen whales are using this habitat. The consistent presence of many baleen whale species suggests that seasonal management actions, such as Seasonal Management Areas near ports that restrict vessel speed or seasonal exclusions associated with offshore construction activity that limit the loudest actions to certain months, may be insufficient to mitigate impacts to these protected species.

## 4.1 Baleen whale ecological patterns

Right whales were most frequently acoustically detected in the NY Bight from fall through spring, with presence >5 days/week for most of that period across years. Right whales were detected least frequently during summer months. The U.S. Mid-Atlantic and the NY Bight are

considered part of the right whale migratory corridor [84, 85], and while is it unclear exactly how right whales use the NY Bight, their extended acoustic presence in the NY Bight during all months of the year suggests that they may not be solely migrating through this region. Similar patterns of extended seasonal occurrence of right whales outside of migration periods have been observed in other areas along their migration corridor [27, 37–39, 88]. Right whales have been observed feeding as far south as Nantucket Shoals [89], and Zoidis et al. [19] observed one instance of a feeding right whale in the NY Bight during the NYSDEC aerial surveys that were contemporaneous with our passive acoustic survey. Right whales were detected predominantly near the shelf edge during spring months, likely during their northward migration to feeding grounds in the Gulf of Maine, and closer to shore during fall months when pregnant females transit southward to calving grounds in southeastern US waters [84, 85]. These spatial patterns corroborate those from the visual surveys conducted by Zoidis et al. [19]. However, right whales were not visually observed during summer months, while they were acoustically detected during June and July in these data and by Murray et al. [16], illustrating the important data gap PAM can fill. Muirhead et al. [12] also described a similar spatial pattern in seasonal detections in the NY Bight in 2008. Detection range estimates indicate right whale presence within and around the high-use Nantucket-Ambrose and Ambrose-Hudson Traffic Separation Schemes.

Humpback whales were among the more frequently detected baleen whale species throughout the NY Bight, with year-round acoustic presence that increased in winter and spring. Both humpback whale song and non-song calls were detected in the NY Bight indicating a range of humpback behaviors occurring in this region including advertisement and foraging [83, 90]. Humpback whale songs have regularly been detected in feeding grounds in the Western North Atlantic Ocean [69, 91–93], and it is thought that singing in feeding grounds may precede and follow migratory movements [94].

Fin whales were acoustically detected on nearly every day in the NY Bight, which is consistent with previous observations in the mid-Atlantic and Gulf of Maine [12, 26, 95], and in the NY Bight in 2008–2009 [12]. The 2008–2009 passive acoustic data show the fewest acoustic detections of fin whales in the fall and highest in winter in the NY Bight, while summer months were not surveyed [12]. Our survey had fewer detections in spring and more in fall and winter. Fin whales had higher detection rates at offshore sites, which may be in part due to the long-range propagation distance of fin whale song [71, 96, 97] originating off the continental shelf. However, the spatial pattern in fin whale acoustic detections observed in this survey suggests that fin whales were within the NY Bight, supported by visual observations during concurrent aerial surveys [19].

The NY Bight likely represents the southerly extent of the range for most sei whales [98], though a low incidence of acoustic detection has been found farther south [26]. Our data present one of the most extensive time-series records of sei whale acoustic occurrence collected to date. All three years of survey data show that March through mid-June is the peak detection period for sei whales in the NY Bight, though sei whales were detected during each season. Sei whales have only recently been the focus of PAM surveys [26, 56, 57, 99], and data on their occurrence in the Gulf of Maine and mid-Atlantic have only recently been analyzed [26]. Davis et al. [26] reported similar seasonal occurrence patterns of sei whales in 2008–2009. Sei whales were detected mostly at sites near the shelf edge, consistent with concurrent visual observations [19]. The TPR of the sei whale detector (0.33) suggests that more than half of the downsweeps were not detected; however, sei whales were detected on more days throughout the year during the acoustic survey than in visual surveys [19], illustrating that PAM methods are beneficial for long-term monitoring of sei whale presence in the NY Bight. For comparison, visual sightings of sei whales were rare between 2017 and 2020 (2 sightings) and occurred

only in April, while sei whales were acoustically detected during every season, with a peak in spring months. Considering the estimated short detection ranges for sei whale downsweeps and the superior detector performance on downsweeps that were not mode dispersed (i.e., the source was close to the site), detected downsweeps were most likely produced by sei whales near the recording sites. We do not suspect that the rate of missed downsweeps by the detector was biased by time, in which case these data represent broad temporal trends of vocalizing sei whale occurrence in the NY Bight, which align with trends observed in the 2008–2009 acoustic data [26].

Blue whales were detected least frequently among the focal species, consistent with the 2008–2009 survey [12, 26] and visual surveys [19]. Future studies should investigate how far into the NY Offshore Planning area blue whales are travelling. Given the long detection range of blue whale song and spatial pattern of acoustic detections, it is likely that most of the detected blue whales in this survey were off the shelf or along the shelf edge. During concurrent aerial surveys, the only blue whale observations were on the continental shelf and at the shelf edge, suggesting that blue whales may not utilize the NY Bight waters. A single blue whale was acoustically tracked by Muirhead et al. [12] on the shelf, which may indicate that individuals transit through the planning area. Blue whales elsewhere in the western North Atlantic have been acoustically detected more off the continental shelf or near the shelf edge than in waters closer to shore [26].

### 4.2 Ambient noise and detection range estimates

At the northeast convergence of the two shipping lanes, site 8A consistently recorded the highest noise levels. Sites adjacent to 8A, where large vessels reduce speed or anchor, recorded lower noise levels. Considering the placement of the recording sites along two major shipping lanes (Fig 1), vessel noise is likely the most dominant contributor to frequencies between 50 and 500 Hz throughout the recording area. Elevated ambient noise levels near shipping lanes can limit the acoustic detection range of low frequency ($< 1$ kHz) baleen whale signals to less than 10 km and potentially restrict their conspecific communication space through increased acoustic masking [43, 67, 82]. We found that noise levels varied by season, which may affect communication space between seasons, but did not differ greatly between survey years.

While our detection range estimates merely provide approximate areas of acoustic coverage for each site and target species during this survey, they were similar to estimates in nearby regions of the western North Atlantic continental shelf [20, 56, 82], with the exception of fin whale detection range estimates. Fin whale detection ranges in the NY Bight were estimated to exceed 150 km from the source, while in Massachusetts Bay, fin whale detection ranges were estimated to be $< 50$ km [82]. We note that our estimated detection ranges for these species are based on noise conditions along the two major shipping lanes of the NY Bight at the time of this survey and are not representative of other regions within the NY Bight that are farther from the noise of the shipping lanes, nor of other regions along the North Atlantic continental shelf. There are several parameters in the model which can vary in real-world conditions (e.g., source level, source depth, transmission loss), as mentioned by Clark et al. [43], and these estimates are intended to demonstrate the potential ranges to which a target signal may be recorded at the sensors in NY Bight during this survey.

### 4.3 Implications

With the recent intensive, multi-modal, marine mammal survey efforts in the NY Bight, this region has gone from being a significant data gap to now one of the most intensively studied regions along the U.S. Atlantic EEZ. Each of these survey methods–systematic or opportunistic

visual surveys, eDNA, and passive acoustics–has advantages and limitations, but when combined, they can yield a more comprehensive understanding of the ecology of cryptic taxa. Visual surveys have long been the most common survey tool for monitoring the presence and density of marine mammals at sea. Visual surveys [18, 19, 100] provide data including species identification, the number of different individuals present, geographic location, and potentially behavioral information, yet are difficult to conduct at night or in other low visibility conditions. Opportunistic survey data [15] can be useful for establishing the presence of protected species, but cannot reliably demonstrate absence of animals in this area due to the lack of systematic effort. eDNA is an emerging survey method with tremendous promise of single or multi-species detection [e.g., 101–103], including marine mammals [14, 104], though questions of detection probability remain [105, 106]. Passive acoustic monitoring is a relatively low-cost way to increase the spatial and temporal survey coverage for marine mammals that generate sounds; it allows for continuous surveillance of the species of interest, providing coverage at night and in poor weather conditions, when visual surveys from aerial or shipboard platforms are not feasible [22], and allows for detection of animals below the surface. Extensive aerial and now passive acoustic survey efforts for monitoring whale occurrence in the NY Bight provides new insights into their seasonal and spatial presence and can help inform more effective conservation and management strategies to minimize impacts to these protected species, yet allow for sustainable marine spatial planning and human use of this region. Integrated analysis of different survey methods [e.g., 20, 107] may provide further opportunities to optimize large scale sustained monitoring efforts for protected species to guide sustainable marine spatial planning in areas of high human use and biological importance.

## Supporting information

**S1 Table. Sensor deployment information for each site and survey year.**
(PDF)

**S2 Table. Daily presence for each focal whale species by month across all sites, and the corresponding percentage of days in which each species was detected.**
(PDF)

**S3 Table. Summary of daily detections for each baleen whale species total days refers to the number of days that were sampled during the 3-year survey.**
(PDF)

**S4 Table. Mean and standard error (Std Error) detection range estimates for each species and site averaged across Year 1, Year 2, and Year 3 for the lowest (5th percentile), median (50th percentile), and highest (95th percentile) noise conditions between 16 October 2017 and 15 October 2020 in the NY Bight.** Blank cells indicate that the estimated detection range exceeded 500 km. Blank cells in the standard error column indicate that there was only one value used for the estimate.
(PDF)

**S5 Table. Daily acoustic detection of North Atlantic right whale (RI), humpback whale (HB), fin whale (FP), sei whale (SE), and blue whale (BL) signals by site in the New York Bight between October 2017 and October 2020.** Presence = 1 denotes a detection of the target species on that day and site, Presence = 0 denotes no detection of the target species, and a blank cell denotes there are no data for that date and site. Data Gap indicates dates with sound recording for a site (0) and dates without sound recording for the site (1).e.
(PDF)

**S1 Fig.** Spectrogram of sei whale downsweeps exhibiting frequency dispersion: A) a sei whale downsweep similar to the templates used for the template detector, B) a multi-path (distorted) sei whale downsweep, which was often not detected by the template detector. Spectrogram was created with a window size = 2048, DFT = 4096, with frequency and time bins of 1.22 Hz and 0.0614 s, respectively.
(TIF)

**S2 Fig. Quantile boxplots of weekly acoustic presence of North Atlantic right whales per site in New York Bight between October 2017 and October 2020, shown as proportion of recorded days per week with confirmed right whale upcall detections across all sensors (purple).** Grey bars indicate the mean number of days per week without data, along the inverted secondary x-axis.
(TIF)

**S3 Fig. Quantile boxplots of weekly acoustic presence of humpback whales per site in New York Bight between October 2017 and October 2020, shown as proportion of recorded days per week with confirmed humpback whale song and non-song detections across all sensors (green).** Grey bars indicate the mean number of days per week without data, along the inverted secondary x-axis.
(TIF)

**S4 Fig. Quantile boxplots of weekly acoustic presence of fin whales per site in New York Bight between October 2017 and October 2020, shown as proportion of recorded days per week with confirmed fin whale 20-Hz pulse detections across all sensors (red).** Grey bars indicate the mean number of days per week without data, along the inverted secondary x-axis.
(TIF)

**S5 Fig. Quantile boxplots of weekly acoustic presence of sei whales per site in New York Bight between October 2017 and October 2020, shown as proportion of recorded days per week with confirmed sei whale downsweep detections across all sensors (orange).** Grey bars indicate the mean number of days per week without data, along the inverted secondary x-axis.
(TIF)

**S6 Fig. Quantile boxplots of weekly acoustic presence of blue whales per site in New York Bight between October 2017 and October 2020, shown as proportion of recorded days per week with confirmed blue whale song detections across all sensors (blue).** Grey bars indicate the mean number of days per week without data, along the inverted secondary x-axis.
(TIF)

**S7 Fig. Boxplots of noise $L_{eq}$ (dB re 1μPa) measurements for the right whale frequency band (70.8–224 Hz), survey year (Year 1 = October 2017–2018, Year 2 = October 2018–2019, Year 3 = October 2019–2020), season and site.**
(TIF)

**S8 Fig. Boxplots of noise $L_{eq}$ (dB re 1μPa) measurements for the humpback whale frequency band (17.8–708 Hz), survey year (Year 1 = October 2017–2018, Year 2 = October 2018–2019, Year 3 = October 2019–2020), season and site.**
(TIF)

**S9 Fig. Boxplots of noise $L_{eq}$ (dB re 1μPa) measurements for the fin whale frequency band (17.8–28.2 Hz), survey year (Year 1 = October 2017–2018, Year 2 = October 2018–2019, Year 3 = October 2019–2020), season and site.**
(TIF)

**S10 Fig. Boxplots of noise L$_{eq}$ (dB re 1μPa) measurements for the sei whale frequency band (44.7–112 Hz), survey year (Year 1 = October 2017–2018, Year 2 = October 2018–2019, Year 3 = October 2019–2020), season and site.**
(TIF)

**S11 Fig. Boxplots of noise L$_{eq}$ (dB re 1μPa) measurements for the blue whale frequency band (14.1–22.4 Hz), survey year (Year 1 = October 2017–2018, Year 2 = October 2018–2019, Year 3 = October 2019–2020), season and site.**
(TIF)

**S12 Fig. Boxplots of noise L$_{eq}$ (dB re 1μPa) measurements for the full frequency band (8.9–2239.6 Hz), survey year (Year 1 = October 2017–2018, Year 2 = October 2018–2019, Year 3 = October 2019–2020), season and site.**
(TIF)

## Acknowledgments

This work could not have been completed without major contributions from several key individuals. Special thanks to Captain Fred Channel and Derek Jaskula for deploying and recovering recording devices in a wide range of seafaring conditions, to Deborah Cipolla-Dennis, Linda Harris, Edward Moore, Tish Klein, and Holger Klinck for invaluable administrative support, to Christopher Tessaglia-Hymes and Raymond Mack for engineering support, to Peter Dugan for optimizing and running automated detection algorithms, to Michael Pitzrick for software support, to Dimitri Ponirakis for ambient noise and detection range software development and support, and to Christopher Pelkie for assisting with data management. We thank JASCO Applied Sciences for AMAR and AURAL support.

## Author Contributions

**Conceptualization:** Lisa A. Bonacci-Sullivan, Meghan E. Rickard, Matthew D. Schlesinger, Susan E. Parks, Aaron N. Rice.

**Data curation:** Bobbi J. Estabrook, Kristin B. Hodge, Daniel P. Salisbury.

**Formal analysis:** Bobbi J. Estabrook, Danielle V. Harris, Kristin B. Hodge, Ashakur Rahaman, Daniel P. Salisbury, Julia M. Zeh, Aaron N. Rice.

**Funding acquisition:** Lisa A. Bonacci-Sullivan, Meghan E. Rickard, Matthew D. Schlesinger, Susan E. Parks, Aaron N. Rice.

**Investigation:** Bobbi J. Estabrook, Danielle V. Harris, Aaron N. Rice.

**Methodology:** Bobbi J. Estabrook, Danielle V. Harris, Susan E. Parks, Aaron N. Rice.

**Project administration:** Lisa A. Bonacci-Sullivan, Meghan E. Rickard, Susan E. Parks, Aaron N. Rice.

**Resources:** Susan E. Parks, Aaron N. Rice.

**Software:** Bobbi J. Estabrook.

**Supervision:** Bobbi J. Estabrook, Susan E. Parks, Aaron N. Rice.

**Validation:** Bobbi J. Estabrook, Danielle V. Harris, Kristin B. Hodge, Ashakur Rahaman, Daniel P. Salisbury, Aaron N. Rice.

**Visualization:** Bobbi J. Estabrook, Aaron N. Rice.

**Writing – original draft:** Bobbi J. Estabrook, Julia M. Zeh, Susan E. Parks, Aaron N. Rice.

**Writing – review & editing:** Bobbi J. Estabrook, Lisa A. Bonacci-Sullivan, Danielle V. Harris, Kristin B. Hodge, Ashakur Rahaman, Meghan E. Rickard, Daniel P. Salisbury, Matthew D. Schlesinger, Julia M. Zeh, Susan E. Parks, Aaron N. Rice.

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
