## [Decision Letter · Decision Letter 0]

15 May 2024

PONE-D-24-13777Passive acoustic monitoring of baleen whale seasonal presence across the New York BightPLOS ONE

Dear Dr. Parks,

Thank you for submitting your manuscript to PLOS ONE. After careful consideration, we feel that it has merit but does not fully meet PLOS ONE’s publication criteria as it currently stands. Therefore, we invite you to submit a revised version of the manuscript that addresses the points raised during the review process.

We look forward to receiving your revised manuscript.

Kind regards,

Vitor Hugo Rodrigues Paiva, Ph.D.

Academic Editor

PLOS ONE

Journal Requirements:

   "Funding for this study was provided by a contract from the New York State Department of Environmental Conservation (https://dec.ny.gov/) to ANR and SEP (Contract C009925). Scientists from NYSDEC (LAB-S, MER) and the NY State Natural Heritage Program (MDS) were involved in survey design and defining project goals, but did not have any influence over the interpretation or presentation of results."

3. For studies involving third-party data, we encourage authors to share any data specific to their analyses that they can legally distribute. PLOS recognizes, however, that authors may be using third-party data they do not have the rights to share. When third-party data cannot be publicly shared, authors must provide all information necessary for interested researchers to apply to gain access to the data. (https://journals.plos.org/plosone/s/data-availability#loc-acceptable-data-access-restrictions) 

4. We note that Figure 1 in your submission contain map/satellite images which may be copyrighted. All PLOS content is published under the Creative Commons Attribution License (CC BY 4.0), which means that the manuscript, images, and Supporting Information files will be freely available online, and any third party is permitted to access, download, copy, distribute, and use these materials in any way, even commercially, with proper attribution. For these reasons, we cannot publish previously copyrighted maps or satellite images created using proprietary data, such as Google software (Google Maps, Street View, and Earth). For more information, see our copyright guidelines: http://journals.plos.org/plosone/s/licenses-and-copyright.

Reviewers' comments:

Reviewer's Responses to Questions

**Comments to the Author**

1. Is the manuscript technically sound, and do the data support the conclusions?

Reviewer #1: Yes

Reviewer #2: Yes

2. Has the statistical analysis been performed appropriately and rigorously? 

Reviewer #1: Yes

Reviewer #2: Yes

3. Have the authors made all data underlying the findings in their manuscript fully available?

Reviewer #1: No

Reviewer #2: Yes

4. Is the manuscript presented in an intelligible fashion and written in standard English?

Reviewer #1: No

Reviewer #2: Yes

5. Review Comments to the Author

Reviewer #1: t would be helpful if the introduction described the New York Bight within the context of the target species’ distributions, seasonal movements, and other activities. These topics are addressed in the discussion, but not the introduction. For example, the introduction does not explain hypotheses related to migration versus residence of some species.

lines 54, 56. Habitat is a species-specific construct that is not synonymous with ecosystem type. Habitat contains the elements necessary for a given species to survive and reproduce. In this case, “ecosystems” likely would be the correct word.

line 60. What is a “survey effort”? Why not simply “surveys”?

line 62. Why is fair in quotation marks? I’m assuming it’s a quotation from the citation, but the meaning is ambiguous. Can you explain in a manner that doesn’t require quotation marks?

line 62. What is a “monitoring survey”? I know what monitoring is, and I know what a survey is, but not what a monitoring survey is.

line 77. Unless these aerial surveys accounted for detection probability, it may not be entirely accurate to say that they identified occurrence patterns.

line 92. PAM is not necessarily synonymous with long-term or continuous data collection.

line 102. Why these five species?

line 105. Neither a monitoring plan nor monitoring can mitigate effects of human activities. Data from monitoring might inform development of actions to mitigate those effects.

It doesn’t seem that there is a research question. That’s okay—surveys were conducted to document species presence—but I’m uncertain that reporting on presence warrants publication in the journal.

lines 111-112. Why these two particular devices? To what extent can the data collected by these devices be integrated, and are there any trade-offs to doing so?

line 134. Why is it necessary or sensible to avoid these areas? Are vessels likely to damage the sensors, for example?

line 155. What was that second verification process?

line 177. It seems unlikely that 25% and 100% yielded identical results. Please elaborate the meaning of “the same results.”

line 190. There is a reference to daily presence here (sei whales) and at line 156 (North Atlantic right whales). What was the temporal resolution of detections of the other species?

line 194. I’m not entirely following. Do you mean that you sought sei whale downsweeps in your data, selected six, used those downsweeps to develop a classifier, and then applied the classifier to the complete recordings?

line 198. Please define down-sampled.

line 209. Why is groundtruthed in quotation marks?

line 225. The multi-modal dispersion question should be described above in the section on sei whales.

line 239. Is “daily detection per site” an average across sites, or was site treated as the sample unit? line 244 appears to imply the latter, but I’m not sure.

line 249. Did you set a threshold AIC difference (e.g., 2 or 4) for considering the strength of evidence for two models to be different?

line 252. This sentence is constructed poorly. Presence is not using R.

line 277. Why is full band in quotation marks?

The results are quantitative, but largely descriptive. What is one to infer biologically from the extent to which the probability of detection of a given species varied among years?

The discussion seems a bit ad hoc. I felt that some of the material on each species could be moved to the introduction to better establish biological questions that the data potentially could answer. Similarly, questions about detection ranges largely were examined in the discussion without context earlier in the manuscript.

Reviewer #2: This is a good paper, I just have a few comments that can be addressed relatively easily. The two larger points is that I think it would be good to include more context on the limitations of PAM and also the findings relating to the interactions between whales, you focus on each as a separate entity without any discussion of hose the presence of one group impacts the presence of others and there is nothing on overlap and interactions. I also think you need to spend more time in the discussion talking about policy implications and future plans to protect these species.

6. PLOS authors have the option to publish the peer review history of their article (what does this mean?). If published, this will include your full peer review and any attached files.

Reviewer #1: No

Reviewer #2: No

---

## [Author Response · Author response to Decision Letter 0]

4 Sep 2024

Response to Reviewers (reviewer comments in plain text, and our replies are in italics)

Reviewer #1

It would be helpful if the introduction described the New York Bight within the context of the target species’ distributions, seasonal movements, and other activities. These topics are addressed in the discussion, but not the introduction. For example, the introduction does not explain hypotheses related to migration versus residence of some species.

We appreciate the suggestion. We added text as suggested to the introduction. 

lines 54, 56. Habitat is a species-specific construct that is not synonymous with ecosystem type. Habitat contains the elements necessary for a given species to survive and reproduce. In this case, “ecosystems” likely would be the correct word.

Thanks for suggesting this. We changed the language accordingly.

line 60. What is a “survey effort”? Why not simply “surveys”?

The reviewer is correct, this should be “surveys”. We changed the text.

line 62. Why is fair in quotation marks? I’m assuming it’s a quotation from the citation, but the meaning is ambiguous. Can you explain in a manner that doesn’t require quotation marks?

We replaced “fair” with “intermittent” to make the sentence less ambiguous.

line 62. What is a “monitoring survey”? I know what monitoring is, and I know what a survey is, but not what a monitoring survey is.

“Passive acoustic monitoring” refers to the method of data collection (also referred to as PAM in the manuscript). These data are the result of a PAM survey. Identical terminology has been used for several other passive acoustic monitoring surveys by other research groups.

line 77. Unless these aerial surveys accounted for detection probability, it may not be entirely accurate to say that they identified occurrence patterns.

We respectfully disagree, since occurrence pattern” can be a series of positive detections or observations in space or time, with no inference for population size, number of animals, or density. 

line 92. PAM is not necessarily synonymous with long-term or continuous data collection.

Fair point. We clarified this in the text.

line 102. Why these five species?

These baleen whales were identified as “species of conservation need” by New York State, and are the likely baleen whale species to be found on the continental shelf within the New York Bight. We added this text and relevant citations to the intro.

line 105. Neither a monitoring plan nor monitoring can mitigate effects of human activities. Data from monitoring might inform development of actions to mitigate those effects.

We appreciate the important clarification. We modified the text accordingly. Thank you for flagging this.

It doesn’t seem that there is a research question. That’s okay—surveys were conducted to document species presence—but I’m uncertain that reporting on presence warrants publication in the journal.

This is particularly helpful feedback. The text was reframed to demonstrate the research question and the overarching conservation needs. 

lines 111-112. Why these two particular devices? To what extent can the data collected by these devices be integrated, and are there any trade-offs to doing so?

We clarified this in the text at the end of the paragraph. Thank you for highlighting this.

line 134. Why is it necessary or sensible to avoid these areas? Are vessels likely to damage the sensors, for example?

“Areas to be avoided” is a U.S. Coast Guard designation for particular waterways for commercial vessels to avoid due to compromised navigation or potential for ship collisions (see https://www.noaa.gov/gc-international-section/marine-protected-areas-mpas-areas-to-be-avoided). We reworded this sentence to make it clearer.

line 155. What was that second verification process?

The second verification process was a review by a second expert human analyst to reduce the potential for single-observer bias. We clarified this in the text.

line 177. It seems unlikely that 25% and 100% yielded identical results. Please elaborate the meaning of “the same results.”

We reworded this sentence for clarification.

line 190. There is a reference to daily presence here (sei whales) and at line 156 (North Atlantic right whales). What was the temporal resolution of detections of the other species?

Thanks for catching this, and it was an accidental omission. Temporal analysis resolution was daily for all species, and we clarified this for all species.

line 194. I’m not entirely following. Do you mean that you sought sei whale downsweeps in your data, selected six, used those downsweeps to develop a classifier, and then applied the classifier to the complete recordings?

Yes. We reworded the sentence to make this clearer.

line 198. Please define down-sampled.

We reworded this sentence and added a definition.

line 209. Why is groundtruthed in quotation marks?

The reviewer is correct that quotation marks are not necessary, and we removed them accordingly.

line 225. The multi-modal dispersion question should be described above in the section on sei whales.

Good suggestion. We moved this sentence accordingly

line 239. Is “daily detection per site” an average across sites, or was site treated as the sample unit? line 244 appears to imply the latter, but I’m not sure.

Thanks for pointing this out. Day was the sample unit, and we clarified the phrasing to make the text in line 244 to make this clearer

line 249. Did you set a threshold AIC difference (e.g., 2 or 4) for considering the strength of evidence for two models to be different?

We set a threshold for the difference in AIC to 2. We added that information to this line.

line 252. This sentence is constructed poorly. Presence is not using R.

We reworded the sentence

line 277. Why is full band in quotation marks?

This sentence was modified to make it a bit clearer and rephrased such that the quotations are not necessary

The results are quantitative, but largely descriptive. What is one to infer biologically from the extent to which the probability of detection of a given species varied among years?

Inferring detection probability of mysticetes through PAM has a number of confounding factors not accounted for here (i.e. vocalization rate), and thus we tried to simplify the summary of our results through describing spatial and temporal patterns.

The discussion seems a bit ad hoc. I felt that some of the material on each species could be moved to the introduction to better establish biological questions that the data potentially could answer. Similarly, questions about detection ranges largely were examined in the discussion without context earlier in the manuscript.

Thanks for the feedback. We modified the discussion accordingly.

Reviewer #2: 

This is a good paper, I just have a few comments that can be addressed relatively easily. The two larger points is that I think it would be good to include more context on the limitations of PAM 

We added a bit more context on limitations of PAM, though we also included citations to references that address PAM limitations in greater detail than we have space to do here.

and also the findings relating to the interactions between whales, you focus on each as a separate entity without any discussion of hose the presence of one group impacts the presence of others and there is nothing on overlap and interactions. 

Excellent suggestion. We added a bit of language to this point in the discussion. However, in terms of regulatory context, each of these species is managed individually without any context of possible interaction, so we didn’t want to get too much into mysticete community composition based on some of the inference limitations from our data.

I also think you need to spend more time in the discussion talking about policy implications and future plans to protect these species.

Helpful suggestion. We added language to the discussion on these topics.

You don’t need to include keywords if they are in your title

Good point. We removed “baleen whale” from the keywords.

In what sense the ecological and economic importance needs to be unpacked.

We added more details about this in the introduction.

This stands in opposition to what you say above and also below – this is a very well surveyed area

We clarified our statement earlier to reiterate that there have been relatively few cetacean surveys, compared to other systematic surveys in the New York Bight. In this section, we are not stating that cetaceans have not been studied in the New York Bight, rather there have not been many surveys conducted in this region.

Please elaborate on the plans

We added context about the planned offshore wind development in the region

It would be good if you discussed the difficulty with this

Describing density and abundance estimation and their limitations goes beyond the scope of this paper since we do not estimate density or abundance with these data.

Please cite https://onlinelibrary.wiley.com/journal/17487692 & https://onlinelibrary.wiley.com/doi/abs/10.1111/brv.12969

We included the second one, but the first link is just the landing page for Marine Mammal Science. We are happy to include more

or will?

Yes, we agree. We changed “can” to “will”.

Why 2m and 1m suspension?

The length of the suspended unit from the seafloor varied by recording device, based on specific equipment configuration of the device.

Please explain more here about which software you used and when visual or aural detection was used

The software used for visual and aural detection is described in the text below this sentence in the paragraph.

This seems like an important finding that should be in the abstract.

We agree. We added some text in the abstract to convey this finding.

---

## [Decision Letter · Decision Letter 1]

1 Oct 2024

PONE-D-24-13777R1Passive acoustic monitoring of baleen whale seasonal presence across the New York BightPLOS ONE

Dear Dr. Parks,

Thank you for submitting your manuscript to PLOS ONE. After careful consideration, we feel that it has merit but does not fully meet PLOS ONE’s publication criteria as it currently stands. Therefore, we invite you to submit a revised version of the manuscript that addresses the points raised during the review process.

We look forward to receiving your revised manuscript.

Kind regards,

Vitor Hugo Rodrigues Paiva, Ph.D.

Academic Editor

PLOS ONE

Journal Requirements:

Reviewers' comments:

Reviewer's Responses to Questions

**Comments to the Author**

1. If the authors have adequately addressed your comments raised in a previous round of review and you feel that this manuscript is now acceptable for publication, you may indicate that here to bypass the “Comments to the Author” section, enter your conflict of interest statement in the “Confidential to Editor” section, and submit your "Accept" recommendation.

Reviewer #1: All comments have been addressed

Reviewer #3: (No Response)

2. Is the manuscript technically sound, and do the data support the conclusions?

Reviewer #1: (No Response)

Reviewer #3: Partly

3. Has the statistical analysis been performed appropriately and rigorously? 

Reviewer #1: (No Response)

Reviewer #3: Yes

4. Have the authors made all data underlying the findings in their manuscript fully available?

Reviewer #1: (No Response)

Reviewer #3: Yes

5. Is the manuscript presented in an intelligible fashion and written in standard English?

Reviewer #1: (No Response)

Reviewer #3: Yes

6. Review Comments to the Author

Reviewer #1: (No Response)

Reviewer #3: I recognize that you have already done a round of reviews and, although I was not one of the first reviewers, it is clear to me that many improvements were made in response to the initial reviewer's comments. The majority of this manuscript looks ready to go to me. However, the detection range section seems not to have received much attention, and I am a little concerned about the calculations there. I would like to (1) make sure I understand the rationale behind the approach used to estimate detection ranges, and (2) see a bit more discussion of the limitations and assumptions of this method, which I believe to be overly simplistic. I am not recommending that the method change dramatically, but it does need to be clear to readers that this is only a very rough approximation. My detailed comments are below. The most important is related to Equation 1, which I don't understand, as described in more detail below.

Line 116: “part of the limits of their geographic range extent”

Text in bold is a little confusing. Perhaps “at the edge of”?

Line 164: Typo, space needed in “eventsof”

Line 166: Wouldn’t one usually say “automated detector” or “automated detection algorithm”?

Please provide a reference be provided for this algorithm.

Line 192: Extra space between presence and .

Line 295: Why is the species' hearing relevant? Isn't the point of this to quantify your acoustic detection range, not to evaluate their ability to hear?

Line 301/Equation 1

I don't have Urick (84) on hand, but I don't recall seeing this formulation in it or anywhere else previously. Is H the depth of the source or the depth of the water column?

If it's depth of the source, that seems odd to me, wouldn't this give very different answers for the different source depths given in Table 2?

The decision to assume cylindrical spreading usually depends on bottom depth, not source depth.

I recognize that different source depths do affect transmission loss, but that also depends on wavelength, which is not part of this equation.

The R/H part of this equation does not make sense to me. I think a reference or a more detailed explanation is needed, I'm not seeing how you get this from the two propagation references (85 & 86). If H is actually the depth of the water column, a bit more explanation would still be helpful.

I am also surprised that you're finding something close to spherical spreading (17log10(R)) in such shallow water over a flat shelf. How is 17 picked instead of the usual 10log10(R) for cylindrical spreading? I could see this making sense for sensors near the shelf break with sources beyond the shelf break in deep water, but it doesn’t seem to make sense for the sensors close to the coast where detection ranges are estimated to be much less than the shelf width.

Figure 4. I appreciate figures with lots of info packed in, but these maps are really difficult to interpret, particularly for the blue and fin whales.

Showing 5th, 50th, and 95th noise percentiles is too much for one plot once they start overlapping. I suggest showing 50th, and putting 5th and 95th in the supplementary materials.

Site names need to be larger on all panels, or omit the names and put a dot at each site location. Scale bar text is also too small.

Line 435: It would be helpful to remind the reader what kind of dBs we’re talking about (dB re 1 uPa).

Line 522: The detector is biased against distant calls, right, since it is struggling with the dispersed ones? So does that mean that your effective detection range may be less than you estimate?

Line 536: More discussion of the limitations of this detection range estimation approach is needed. It is overly simplistic to use a variant of the sonar equation to estimate detection ranges when using a detector based on what appears to be some kind of computer vision or spectral cross-correlation. The sensitivity of these methods varies considerably, and does not always seem to be clearly related to SNR. Detectability may also depend on signal bandwidth and duration, which differs between species.

Here you're assuming if the RL = ambient noise level, then the call is detectable. Is that a fair assumption? SNR must be quite low in such cases.

If a blue whale is beyond the shelf break, does that signal propagate well up onto the shelf? How much do seasonal changes in oceanography have the potential to influence detectability? If right whales are vocalizing at 2m depth, are they sometimes above the thermocline, and sometimes below, and could that make a difference at these ranges?

Line 548: What is your explanation for this discrepancy? (150km vs < 50km).

Supplementary docx:

The ambient noise box plots are hard to read, the text is overlapping, and the y axes seem to cover and unnecessarily broad range. There don’t seem to be values lower than about 80 or above about 130, so perhaps the range could be reduced to make the plots easier to read.

7. PLOS authors have the option to publish the peer review history of their article (what does this mean?). If published, this will include your full peer review and any attached files.

Reviewer #1: No

Reviewer #3: No

---

## [Author Response · Author response to Decision Letter 1]

15 Nov 2024

Reviewer #3

Line 116: “part of the limits of their geographic range extent”

Text in bold is a little confusing. Perhaps “at the edge of”?

We agree with this suggestion, and changed that part of the sentence to “at the edge of”.

Line 164: Typo, space needed in “eventsof”

We corrected the mistake

Line 166: Wouldn’t one usually say “automated detector” or “automated detection algorithm”?

Please provide a reference be provided for this algorithm.

Yes, that was a mistake – we changed “automated detector algorithm” to “automated detector”. 

We do not have references for these algorithms. We specified that they are custom algorithms in line 161. We also provide detector performance metrics in the supplementary materials.

Line 192: Extra space between presence and .

We removed the space

Line 295: Why is the species' hearing relevant? Isn't the point of this to quantify your acoustic detection range, not to evaluate their ability to hear?

We agree with the reviewer’s point, we removed this portion of the sentence.

Line 301/Equation 1

I don't have Urick (84) on hand, but I don't recall seeing this formulation in it or anywhere else previously. Is H the depth of the source or the depth of the water column?

If it's depth of the source, that seems odd to me, wouldn't this give very different answers for the different source depths given in Table 2?

Thank you for pointing this out. The formula that we from Urick (1983, page 22) is the passive sonar equation. We simplified RL from the original formula, which is derived from Noise Level (NL) – Receiving Directivity Index (DI) + Detection Threshold (DT). We modified the text to include an additional reference (not additional to this manuscript, however) from which these methods followed. 

The Equation 1 references a transmission loss model that we used, which is plugged into the passive sonar question as TL. As stated in the paragraph, H is the depth of the source. Each species has a different estimated detection range value, and therefore, yes, different answers for different source depths. Our detection range estimates are based on the typical depths and frequencies of each target species.

The decision to assume cylindrical spreading usually depends on bottom depth, not source depth.

Our decision for cylindrical spreading was due to bottom depth, not source depth. We removed the misleading text from the sentence on line 301.

I recognize that different source depths do affect transmission loss, but that also depends on wavelength, which is not part of this equation.

However, frequency is the reciprocal of wavelength, and frequency is include in our TL model.

The R/H part of this equation does not make sense to me. I think a reference or a more detailed explanation is needed, I'm not seeing how you get this from the two propagation references (85 & 86). If H is actually the depth of the water column, a bit more explanation would still be helpful.

This approach is based on modeling approaches used in Frankel et al. 2002, and Clark et al. 2009, and we added the citations to the text.

I am also surprised that you're finding something close to spherical spreading (17log10(R)) in such shallow water over a flat shelf. How is 17 picked instead of the usual 10log10(R) for cylindrical spreading? I could see this making sense for sensors near the shelf break with sources beyond the shelf break in deep water, but it doesn’t seem to make sense for the sensors close to the coast where detection ranges are estimated to be much less than the shelf width.

Thank you for the comment. Based on findings from empirical playback experiment in similar coastal habitats along the northern North Atlantic continental shelf, we used 17logR in the transmission loss model. We added these two references to support the use of this value to the manuscript:

Estabrook BJ, Tielens JT, Rahaman A, Ponirakis DW, Clark CW, Rice AN. Dynamic spatiotemporal acoustic occurrence of North Atlantic right whales in the offshore Rhode Island and Massachusetts Wind Energy Areas. Endangered Species Research. 2022;49:115-33.

Hatch LT, Clark CW, Van Parijs SM, Frankel AS, Ponirakis DW. Quantifying Loss of Acoustic Communication Space for Right Whales in and around a U.S. National Marine Sanctuary. Conservation Biology. 2012;26(6):983-94.

Figure 4. I appreciate figures with lots of info packed in, but these maps are really difficult to interpret, particularly for the blue and fin whales.

Showing 5th, 50th, and 95th noise percentiles is too much for one plot once they start overlapping. I suggest showing 50th, and putting 5th and 95th in the supplementary materials.

The 5th percentile ranges were already excluded from the figure for fin and blue whales for exactly this reason. However, we feel that the 3 percentiles for right whales, humpback whales, and sei whales are not confusing. Excluding the 95th percentile ranges from the fin and blue whale plots will not eliminate the significant overlap of the 50th percentiles, and we therefore do not agree with the reviewer that excluding the 95th percentile will benefit the ease of interpretation for this figure.

Site names need to be larger on all panels, or omit the names and put a dot at each site location. Scale bar text is also too small.

We agree that the site labels and scale bar fronts were too small. To help clean up the figure, we removed the site labels completely. We feel that the center point of each circle is enough to denote the recorder locations, which are also presented in Fig 1. We increased the font size of the scale bars.

Line 435: It would be helpful to remind the reader what kind of dBs we’re talking about (dB re 1 uPa).

We added the reference pressure as a reminder

Line 522: The detector is biased against distant calls, right, since it is struggling with the dispersed ones? So does that mean that your effective detection range may be less than you estimate?

We had stated in this section that “detected downsweeps were most likely produced by sei whales near the recording sites”, which we feel makes clear that the reported sei whale detections are likely closer to the recording device than what the recording device would be capable of recording, given the detection range model.

Line 536: More discussion of the limitations of this detection range estimation approach is needed. It is overly simplistic to use a variant of the sonar equation to estimate detection ranges when using a detector based on what appears to be some kind of computer vision or spectral cross-correlation. The sensitivity of these methods varies considerably, and does not always seem to be clearly related to SNR. Detectability may also depend on signal bandwidth and duration, which differs between species.

Here you're assuming if the RL = ambient noise level, then the call is detectable. Is that a fair assumption? SNR must be quite low in such cases.

If a blue whale is beyond the shelf break, does that signal propagate well up onto the shelf? How much do seasonal changes in oceanography have the potential to influence detectability? If right whales are vocalizing at 2m depth, are they sometimes above the thermocline, and sometimes below, and could that make a difference at these ranges?

We added an additional source for the methods we used to estimate the detection range for each species, which describes the model in greater detail, including how signal frequency bands are incorporated in the model. We clarified in the methods and discussion that this model is based on variables whose values can vary greatly, and that this model does not contain all variables possible to more accurately estimate detection ranges. We specified that we present our range estimates as approximate, plausible ranges within which each PAM sensor may be capable of detecting each species to illustrate the potential area of coverage in this survey. We also clarified that these estimates are specific to this study only, for these specific site locations. 

Our assumption for this model is that a target signal can be detectable if the received level exceeds the ambient noise levels, given noise and source level, as well as bathymetry. We state this assumption in the methods section with an added note that variables are variable and these are approximate estimates for the purposes of illustrating potential acoustic coverage around each site location. We acknowledge the assumptions made with this model and specify that seasonal variation is not considered. 

Line 548: What is your explanation for this discrepancy? (150km vs < 50km).

These differences are likely due to differences in slope along the continental shelf, shadowing, depth-related attenuation, but it is also possible that more refined modeling approaches were used in more recent work.

Supplementary docx:

The ambient noise box plots are hard to read, the text is overlapping, and the y axes seem to cover and unnecessarily broad range. There don’t seem to be values lower than about 80 or above about 130, so perhaps the range could be reduced to make the plots easier to read.

There are values below 80 dB in the boxplots (e.g., see S9 Fig, site 09A Year-1), therefore, changing the minimum would result in cutting off the plot. Similarly, some boxplots exceed 140 dB (e.g., see S10 Fig, site 08A, Year-1), and would result in cutting out data if we reduced it as suggested. We fixed the overlapping text along the y-axis.

---

## [Editor Report · Decision Letter 2]

19 Nov 2024

Passive acoustic monitoring of baleen whale seasonal presence across the New York Bight

PONE-D-24-13777R2

Dear Dr. Parks,

We’re pleased to inform you that your manuscript has been judged scientifically suitable for publication and will be formally accepted for publication once it meets all outstanding technical requirements.

Kind regards,

Vitor Hugo Rodrigues Paiva, Ph.D.

Academic Editor

PLOS ONE
---

## [Editor Report · Acceptance letter]

29 Nov 2024

PONE-D-24-13777R2 

PLOS ONE

Dear Dr. Parks, 

I'm pleased to inform you that your manuscript has been deemed suitable for publication in PLOS ONE. Congratulations! Your manuscript is now being handed over to our production team.

Kind regards, 

on behalf of

Dr. Vitor Hugo Rodrigues Paiva 

Academic Editor

PLOS ONE